# Mechanical Properties of Carbon Fiber-Reinforced Plastic with Two Types of Bolted Connections at Low Temperatures

**DOI:** 10.3390/polym16121715

**Published:** 2024-06-16

**Authors:** Hua Li, Feng Guo, Chenglin Han, Wei Su, Shuqi Wen

**Affiliations:** 1School of Aeronautical Engineering, Jilin Institute of Chemical Technology, Jilin 132022, China; guofeng1993@jlict.edu.cn (F.G.); sw@jlict.edu.cn (W.S.); wenshuqi0524@163.com (S.W.); 2Department of Aeronautical Engineering, School of Aerospace Engineering, Shenyang Aerospace University, Shenyang 110136, China; hclcadcam@163.com

**Keywords:** carbon fiber-reinforced plastics, bolted joint, static load tensile, tensile fatigue test

## Abstract

Carbon fiber-reinforced plastic (CFRP) is frequently utilized as a bolted joint material in aircraft applications because of its high specific strength and specific modulus. Therefore, the performance of CFRP under −50° is significant. Here, we discuss the specimens of two bolted connections (single-nailed and double-nailed) used for static load tensile and tensile fatigue tests. We obtained the failure curves and fatigue life relationships of the specimens with two different connection methods at different tightening torques (2 N/m, 4 N/m, and 6 N/m) and low room temperatures. Our analysis reveals the effect of the bolt tightening torque and temperature on the structural mechanical properties of a CFRP bolted joint. It provides a data reference for researchers to design a composite bolted joint structure in an airplane flight environment.

## 1. Introduction

Carbon fiber-reinforced plastic (CFRP) is an excellent structure–function integrated engineering material with good structural integrity and high potential [1,2], and the use of CFRP in aerospace manufacturing is rapidly increasing. In practical applications, the factors affecting the mechanical connection strength of composite materials are far more complex, such as the connection form, connection geometry, connected environment, and so on. All these influencing factors introduce difficulties into the analysis of the mechanical connection strength of CFRP. Therefore, it is important to study the factors influencing the connection strength of composite materials [3,4,5,6].

The cohesion between laminates is an important guarantee for excellent CFRP mechanical properties. Although CFRP has better integrity, due to constraints in the design, preparation, and maintenance, connecting structures are still inevitable in large-scale structures. At present, bolt connections have several advantages, such as good reliability, no limited thickness of the connecting parts, and so on, and are the main form of composite material connection [7,8,9]. The design of composite joint structures is difficult due to the brittle characteristics of the composites and the anisotropy of structure properties [10]. Furthermore, the diversity of composite components and fiber structures, the complex three-dimensional stress and strain distribution in the connection, and the complex damage modes ultimately complicate the design of composite connections.

In addition, mechanical connections do not satisfy ideal elastic or plastic assumptions under load [11]. Bolt joint holes are an obvious shortcoming, because the stress concentration at bolt holes is often the main cause of damage and leads to poor fatigue resistance. Moreover, poorly designed or installed joints not only cause joint failure but also lead to reductions in the durability and reliability of the structure [12,13]. Therefore, it is necessary to conduct research and analysis on the composite materials of a bolt connection, study the influence of material systems and external factors of a bolt connection on the failure mode and the effect of the overall structure, provide an effective input for optimal composite bolt connection structure design, and provide beneficial theoretical and experimental support for the overall structure design [14].

Extrusion failure is the main failure mode in a bolted connection, which is a sudden catastrophic failure mode; the extrusion failure of bolted joints can be summarized as a process of compression damage accumulation. It can be divided into four stages: damage initiation, damage extension, partial fracture, and structural fracture. The main characteristics of extrusion failure include fiber micro-yielding, matrix cracking, delamination, and out-of-plane shear cracking; in addition, extrusion strength and failure modes are related to the lateral restraint (preload force applied to a fastener) and “toughness” of a composite laminate [15,16,17,18]. Riccio et al. [19] studied the damage initiation and extension of bolted structures with different geometric parameters, as well as the ultimate bearing stress and specific strength of joint parameters such as the bolt diameter, hole end distance, screw hole clearance, and bolt joint tightening torque in bolted joints. The results show that, in the absence of bolt joint tightening torque, there is a decrease in the ultimate stress at the bolted joints, as the ratio of the end distance to the hole diameter increases [19,20]. Despite the complexity of the extrusion failure process of composite bolted joints, extrusion failure is mainly caused by the compression failure of the fibers and matrix [21,22,23]. The effect of the CFRP prefabricated fiber-laying method on the structural strength of bolted structures was experimentally investigated, and the results of the study found that alternating layers of fibers improved the strength of the bolted joints more than continuous layers of fibers. Through many experiments, the fiber layer sequence requirement principle was summed up: the fiber 0° layup ratio should account for more than one half, and the 90° layup ratio should be as small as possible. The connection damage mode for the expected local extrusion damage, which can effectively improve the structural strength of the bolt and effectively reduce the load direction, changes as the structure experiences adverse effects [3,24,25]. The results of a large number of experimental studies on composite bolted joints have shown that the fatigue characteristics of bolted joints have a strong relationship with the number of bolts in the joints [5,13,26]; the fatigue life of the joints can be effectively be improved by applying an appropriate tightening torque to the bolted joints, and the tightening torque can effectively inhibit the delamination of the composite laminates in the joints. The fastening element is one reason for the relaxation of the preloading of a bolted connection [4,27,28]; the large assembly gap can be removed by changing the preload force of the fastener, so that the large assembly stress can be generated inside the structure, and the mechanical properties can be enhanced. Nezhad et al. [1] investigated the damage process of CFRP composite bolted joints under quasi-static loading, including both single- and three-bolt types. It was found that the final damage point corresponded to in-layer damage, and the damage size reached the edge of the countersink.

An airplane faces low-temperature environments during high-altitude flight; therefore, the mechanical properties of CFRP as a bolted joint material under low temperatures have attracted much attention. Shindo et al. [29] conducted many experiments; analyzed the tensile, bend, and compressive mechanical properties of composite materials at different temperatures; and summarized the material failure mechanisms. With the continuous decrease in temperature, the mechanical properties of the materials were effectively improved. However, a material’s failure mechanism at different temperatures is different. Similarly, the tensile and molecular behaviors of the resin in CFRP are reflected in its compressive properties [30,31]. The compressive modulus and strength of the resin continue to increase as the temperature decreases, and the material destructive strain decreases. Due to the decrease in the mobility of the polymer chains, the intermolecular bonding at the Young’s modulus and tensile strength of the matrix tend to increase the strength of the material [32,33,34]. According to the time–temperature superposition principle, a lower temperature brings about longer stress relaxation times. Stress relaxation in materials can be completely prevented at low temperatures, leading to an increase in stiffness [35,36,37]. Therefore, it is of great significance to study the mechanical properties of a bolt structure at low temperatures.

In this work, we optimized the CFRP plywood lay-up methods and prepared it using autoclave molding. The as-prepared sheets were cut and perforated to the designed dimensions. The drilled laminates were carefully screened; the specimens without burr and defect damage were selected using zoomed observation of the hole edge, and the treated sheets were bolted together with two types of connections: single and double nails. Static load tensile and tensile fatigue tests were carried out to test the magnitude of the joint strength of the specimens under different conditions. Our study successfully revealed the effect of the tightening torque and temperature on the strength of the connection at the bolted joints. The final purpose of the experiment is to provide a data reference for the design of CFRP bolted structures in the flight environment of the airplane.

## 2. Materials and Methods

### 2.1. Preparation of the CFRP Laminate

The unidirectional carbon fiber/epoxy prepreg (5208/T300) with a thickness of 0.15 mm and a resin volume fraction of 33% was purchased from Guangwei Composites Co(Weihai, China). The CFRP laminate consisted of 16 layers of unidirectional carbon fiber/epoxy prepreg laid by hand in a stacking sequence of [0°/90°] 8s, and the size of each layer was 300 mm × 300 mm × 0.15 mm (Figure 1a). As shown in Figure 1b, the laid prepreg was encapsulated, according to the requirements of autoclave curing.

After completing the laminate lay-up process, the prepared laminate was placed in the autoclave (Figure 2a) and cured under a molding pressure of 0.6 MPa. The laminate was heated from room temperature (25 °C) to 120 °C at a steady rate (1–2 °C/min). The temperature was held at 120 °C for 90 min; then, the laminate was cooled with an opening fan. After cooling the temperature to under 40 °C, the mold was removed, and the cured laminate was cut to the designed size (145 × 36 mm). The pressure accuracy was ±0.01 Mpa, and the temperature accuracy was ±1.1 °C. 

### 2.2. Fabrication of the Bolt Connection Specimen

The as-prepared CFRP laminates were connected with bolts in the next step. An appropriately proportioned end spacing to hole diameter ratio under the bolted joint results in a more uniform transfer of load at the bolted joint, playing a vital role in increasing the load-carrying capacity of the bolted joint. Therefore, the size and position of the drilling holes in the laminate met the following requirements [19]: the ratio of the sample width to the hole diameter was 6:1, and the ratio of the end distance to the aperture was 5:1. Each specimen was composed of three laminated plates with a length of 145 mm, a width of 36 mm, and an aperture of 6 mm, as shown in Figure 3b, where the width of the sample is expressed by W (mm), the hole diameter is expressed by D (mm), and the end distance is expressed by e (mm). The thickness was 2.1 ± 0.05 mm. The double shear ends were filled with an equal thickness of laminates to ensure that the bending of the double shear ends was not adversely affected by the clamping force of the fixture. Afterwards, the as-prepared samples were connected with M6 strength 12.9 alloy steel bolts, tightening with a torque wrench.

### 2.3. The Tensile Test

The tensile strength of the as-prepared bolt-connected specimens was tested, and the tensile fatigue was then tested based on the tensile strength results. The test conditions were set up strictly according to the ASTM D3039/D3039M-17 standard [38], which is widely used in tensile and fatigue testing. According to the geometric specifications of the tensile specimens (Figure 3a,b), the tests were carried out on a universal testing machine with different numbers of bolts, tightening torques, and temperatures. After obtaining the strength of the specimens under different variables in the tensile test, the specimen with the higher strength of the two bolt-connecting methods was selected to test the tensile fatigue. Each test was repeated at least three times to ensure the accuracy of the test results.

We used a format of XPJL-ZZ to differentiate the large number of testing results in different test conditions. The first position X represents the testing temperature; A is room temperature, and B is the low temperature of −50 ℃. The number in the position of P represents the value of the tightening torque. The position of J represents the connection mode, where D is the single-nail connection, and S is the double-nail connection. The position L represents the test mode, where T is the static load tensile test, and V is the fatigue test. In addition, ZZ indicates the specimen number starting from 01.

The extrusion strength is used to compare the impact of the different parameters of each specimen’s bolted connection on the bearing capacity of the connection, which can be obtained from the extrusion stress during failure. The value of the extrusion strength is expressed by Equation (1):(1)$$\sigma_{bt}=F/nDt,$$
where *σ_bt_* denotes the extrusion strength at the bolted connection holes, *F* denotes the maximum breaking load of the bolt-connected specimen, *D* denotes the hole diameter, *n* denotes the number of nail holes in a single row of connection, and *t* denotes the effective thickness of the laminated connection.

The detail of the tests is schematically shown in Figure 4a. The specimens were connected to the test tensile machine with fixtures. Notably, the specimen needed to be kept upright to avoid eccentricity under strain, tightening with the connecting bolt in the direction of the tensile load, as well as the tightening torque, to reduce the slip caused by the connection gap during the test. Vertical stress, at a steady loading rate (2 mm/min), was applied to the fixed specimen. The damage of the specimen was observed and recorded when the tensile machine loaded a single static stretch onto the specimen.

The maximum failure load data of every bolt connection under different conditions were obtained, and the bolt-connection mode with better performance was selected for the following fatigue test by comparing the failure strength of two bolt-connection modes under the same condition. The tensile fatigue test was set with reference to the mean failure load for an optimally bolted joint, with the load coefficients as 85%, 80%, and 75% of the failure strength of the tensile specimens under static load, respectively. The loading waveform is sinusoid with a frequency of 10 Hz, where the horizontal axis is the loading cycles, and the vertical axis is the amplitude of loading strength, as shown in Figure 5. Each group of specimens was subjected to 1,000,000 cycles of fatigue loading, until the specimens were damaged. When the bolt joint was completely damaged or the permanent deformation of the screw hole was more than 1.5 times the diameter of the screw hole, the test specimen was regarded as reaching the fatigue limit. If the test specimen was not damaged after 1,000,000 cycles of loading, the fatigue limit of the specimen was regarded as 1,000,000.

## 3. Results and Discussion

### 3.1. Static Load Tensile Tests

The tensile test data of the load displacement and corresponding failure performance were recorded and are shown in Table 1 and Table 2, and the statistical results of the tests are shown in Figure 6.

As shown in Figure 6a,c, in the early stage of tensile loading, the load of the specimen was positively correlated with the displacement; because of the clearance fit between the bolt and the hole, the initial load strength of the specimen changes little at the beginning. With the load increasing to 80% of the failure load, the specimen produced a fracture sound, and several abrupt inflection points occurred in the load–displacement curve, which were consistent with the theoretical load–deformation phenomena for bolt joint damage [14]. The testing results suggest that the number of bolts is one of the key factors affecting the connection performance of the joint, with the increase in the number of bolts, the stress concentration reduced, resulting in an effective reduction in the failure strength at the screw hole.

Comparing the tensile results between the two connection methods, as shown in Figure 6b,d, the double-nail connection performed more than 20% higher than the single-nail connection in the failure load, and the damage displacement of the double-nail connection was smaller than the single-nail connection as well. This phenomenon has been confirmed in reported works [24,25].

In terms of the results for the same bolted connection methods at different temperatures and tightening torques, as shown in Figure 6e,f, as the tightening torque of the bolt joint increased, the ultimate failure load of the specimen also increased. The result indicates that part of the force in the tightening torque transforms into the compression force of the laminates and strengthens the bonding degree between the laminates, which effectively prevents the local cracking at the screw hole and the stratification of the laminates. That is, the increase in the tightening torque enhances the strength of the bolted connection structure. At the low-temperature condition, the strength of the bolted structure was enhanced as well. The low temperature reduces the internal thermal stress of the material and intermolecular movement speed, which may lead to the closer bonding of the fiber and resin [39,40,41]. 

The failure damage morphology of the specimens is shown in Table 3. The results from the damaged specimens indicate that extrusion failure was the main cause of the structure failure, for both the single-nail and double-nail connections, which is consistent with the results in the reported work [15]. For the damaged parts of the joints of the failed specimens, the fibers at the angles of 0° and 90° showed different formations. In the direction of 90°, the fibers broke with the damage of the specimens, while, in the direction of 0°, they maintained general integrity, just slipping with the direction of the damage load. In addition, the matrix resin exhibited an obvious fragmentation, which suggests that the matrix resin and the 90° direction fiber bear most of the tensile load.

### 3.2. Tensile Fatigue Tests

According to the tensile test data results, the double-nail connection had a higher tensile strength than the single-nail connection. Therefore, the tensile fatigue tests were carried out mainly focusing on the double-nail connected specimens. The overall test data of the pull–tension fatigue tests are shown in Table 4 and Table 5, the fatigue damage effect of the specimen is shown in Table 6, and the statistical results are shown in Figure 7 and Figure 8.

The fatigue life results (Figure 6) and the fatigue life S-N curve (Figure 7) show that the trend of the fatigue curve under the same temperature condition was generally the same and approximately linear. With the decrease in the cyclic stress ratio, the fatigue life of the bolt joint of the composite material showed an increasing trend, and the test results under each stress ratio were less dispersed. At the same temperature, an appropriate increase in the tightening torque enhanced the fatigue life of the specimen [29]. Under the same tightening torque, both the fatigue life and strength at low temperature were better than those at room temperature. The increase in the tightening torque and the decrease in the temperature together improve the fatigue strength and fatigue life of CFRP bolted joints.

The damage morphology of the specimens is shown in Table 6. The compression force transformed by the tightening torque lowered the gap between adjacent laminates, gradually increasing the contact area between the plates, which inhibited the expansion of the extrusion failure hole during the fatigue loading process, inhibited the stress concentration of the laminates, limited the occurrence of lamination and local cracking between the laminates, and enhanced the connection strength and fatigue life of the specimens [27,28]. As for the impact of the temperature, there was a large amount of heat generated on the contact surfaces in the fatigue test due to the repeated impacts on the nail holes. Due to the large gap between the thermal expansion coefficient of the fiber and the matrix, the high temperature may accelerate the chaotic intermolecular movement, diminishing the degree of intermolecular bonding. In that situation, too much thermal stress is generated inside the material and is difficult to release, which results in micro-crack damage inside the material, thereby reducing the mechanical properties. In contrast, the test at a low temperature can effectively reduce the high temperature caused by friction and impacts on the contact surfaces, and the thermal stress can be offset to a certain extent, consequently stabilizing the material performance and increasing the fatigue life [30,31,32,33].

The morphology of the laminate interface at different temperatures is shown in Figure 9a,b. At low temperature, the bond between the resin and the fiber was closer, and the resin content on the fiber was stickier than at room temperature. As for the fracture interface shown in Figure 9c,d, the fiber in the direction of 90° and the resin were damaged, while the fiber preservation degree in the direction of 0° was relatively intact. For the fracture at room temperature, the fractured fibers at 90° existed only in the direction of their own layup, while at low temperature, there were also 90° direction fractured fibers in the 0° direction. This is because the combination between the adjacent layers of the laminate is stronger at low temperature than at room temperature, as mentioned in Section 3.1, resulting in several 90° direction fractured fibers in the 0° direction paved layer.

## 4. Conclusions

A simple and efficient method to prepare CFRP bolted laminates was proposed, and static tensile tests of CFRP bolt-connected structures under different conditions, including the number of bolts, the tightening torque, and the temperature, were carried out. The results indicate that the CFRP bolt-connected structures had mainly extrusion failure with partial shear failure at low temperature and room temperature.

The tensile strength of the double-nail bolted specimen had a more than 20% higher strength than that of the single-nail specimens. The structural strength of the CFRP bolts increased with the increase in the tightening torque of the bolt joints, and the tensile failure load strength increased at low temperature compared with room temperature.

The failure of the bolt structure occurred obviously at the connecting hole. With the increase in the tightening torque and the decrease in the temperature, the static load tensile mechanical properties of the CFRP bolt structure were improved.

The fatigue curves of the bolt-connected structures were less affected by temperature. With the increase in the tightening torque of the bolt joints, the fatigue times of the bolt-connected components under the same fatigue stress ratio increased. The fibers in the 90° direction of the bolted connection broke with the damage of the specimen, while the fibers in the 0° direction remained relatively intact and slipped with the direction of the damage load.

## Figures and Tables

**Figure 1 polymers-16-01715-f001:**
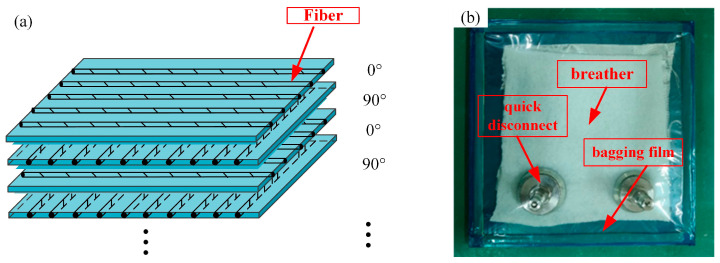
Preparation of laminate: (**a**) laminate paving diagram and (**b**) laminate curing encapsulation.

**Figure 2 polymers-16-01715-f002:**
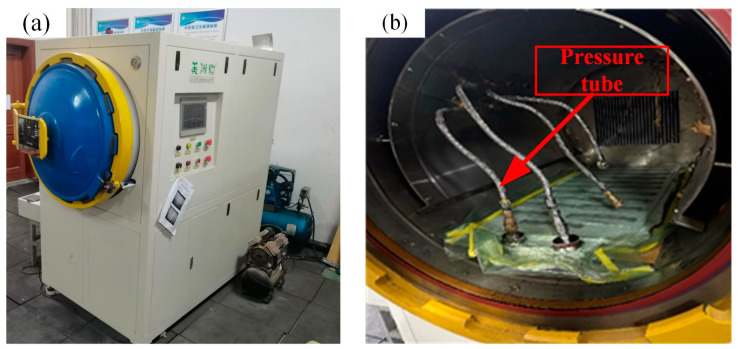
The internal structure of the autoclave for curing: (**a**) the autoclave and (**b**) the construction of the autoclave.

**Figure 3 polymers-16-01715-f003:**
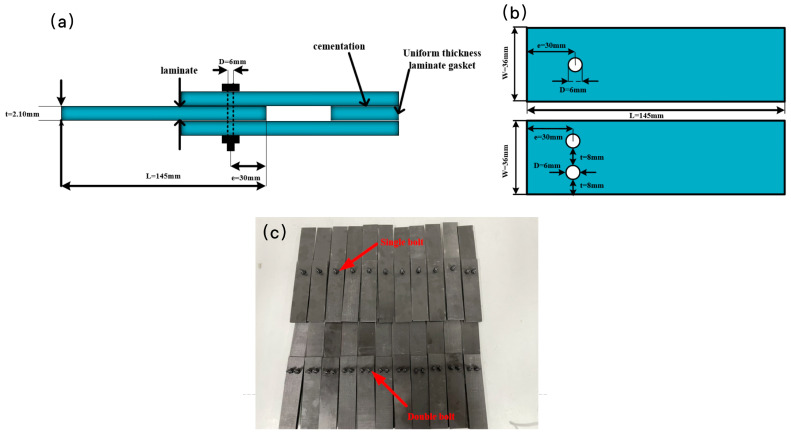
Geometric specifications of the bolt-connected specimens: (**a**) schematic of sample connection mode, (**b**) sample size, and (**c**) photo of real samples.

**Figure 4 polymers-16-01715-f004:**
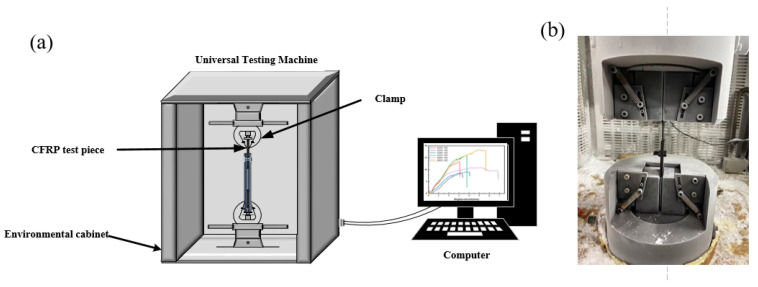
Experimental set-up of tensile test: (**a**) schematic of the tensile machine and (**b**) the photo of clamping mode.

**Figure 5 polymers-16-01715-f005:**
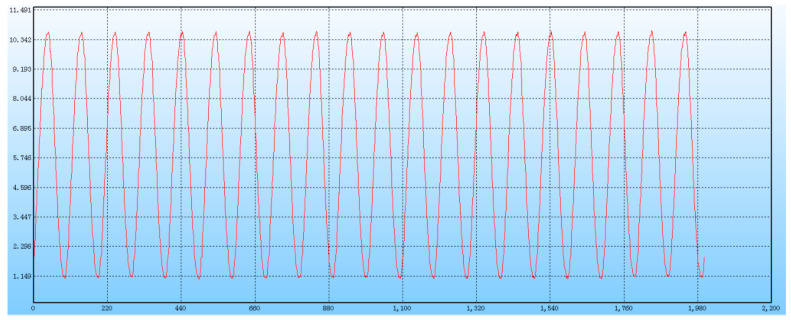
Fatigue loading curve waveform.

**Figure 6 polymers-16-01715-f006:**
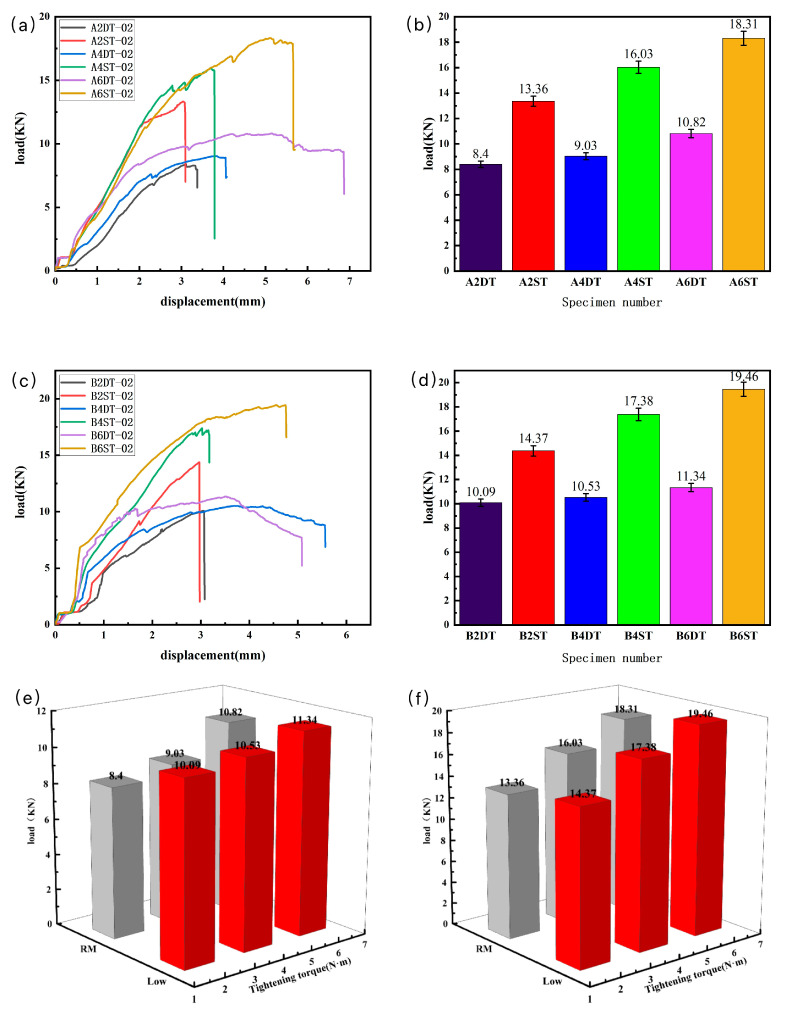
Static load tensile test result: (**a**) load–displacement curve at room temperature, (**b**) average breaking load at room temperature, (**c**) load–displacement curve at low temperature, (**d**) average breaking load at low temperature, (**e**) results of single-nail connection, (**f**) results of double-nail connection.

**Figure 7 polymers-16-01715-f007:**
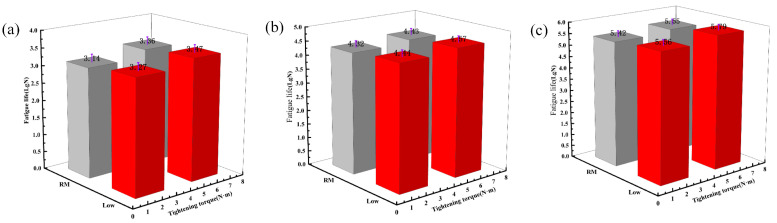
Fatigue test result: (**a**) 85% stress fatigue life results, (**b**) 80% stress fatigue life results, (**c**) 75% stress fatigue life results.

**Figure 8 polymers-16-01715-f008:**
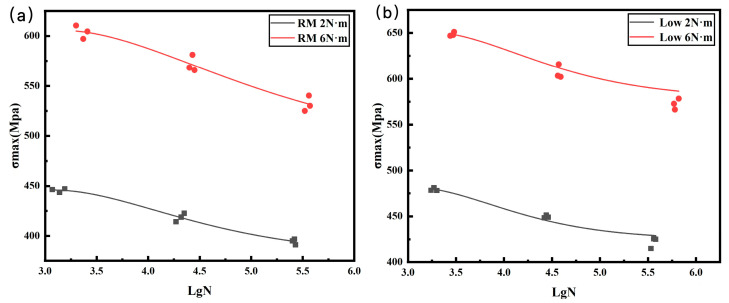
Fatigue life fitting curve: (**a**) room temperature, (**b**) low temperature.

**Figure 9 polymers-16-01715-f009:**
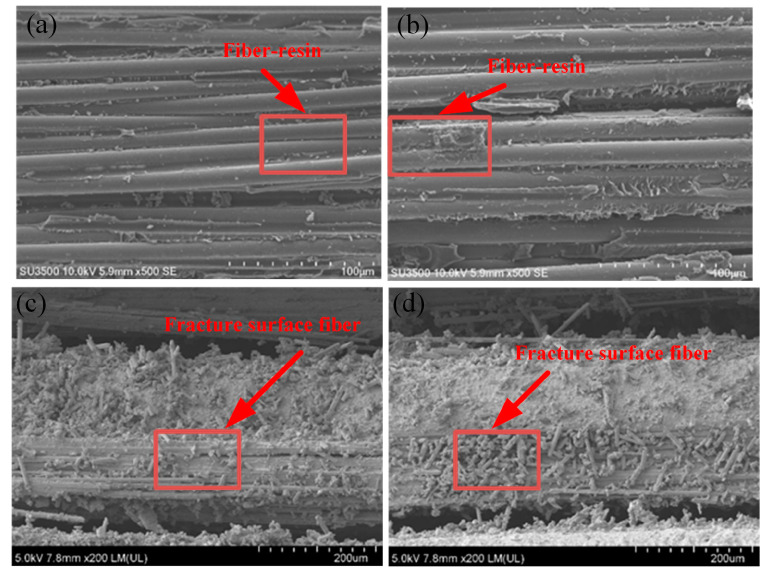
Microscopic morphology of the surface of the specimens: (**a**) room-temperature laminate, (**b**) low-temperature laminate, (**c**) room-temperature fracture surface, and (**d**) low-temperature fracture surface.

**Table 1 polymers-16-01715-t001:** Static load tensile test results at room temperature.

Specimen Number	Maximum Failure Load (KN)	Maximum Load Displacement (mm)	Tensile Strength (MPa)	Average Failure Load (KN)	Average Load Displacement (mm)	Average Strength (MPa)
A2DT-01	8.40	3.11	660.21			
A2DT-02	8.28	3.14	654.96	8.40	3.12	661.29
A2DT-03	8.52	3.12	668.70			
A2ST-01	13.28	2.96	523.61			
A2ST-02	13.33	3.07	534.05	13.36	3.04	532.58
A2ST-03	13.46	3.09	540.07			
A4DT-01	9.00	3.80	707.36			
A4DT-02	9.03	3.83	714.29	9.03	3.83	711.77
A4DT-03	9.05	3.87	713.66			
A4ST-01	15.88	3.68	614.48			
A4ST-02	15.97	3.72	639.82	16.03	3.72	629.19
A4ST-03	16.24	3.75	633.26			
A6DT-01	10.79	5.12	848.05			
A6DT-02	10.82	5.15	855.71	10.82	5.15	857.46
A6DT-03	10.86	5.18	868.61			
A6ST-01	18.25	5.07	713.81			
A6ST-02	18.33	5.10	712.60	18.31	5.09	716.77
A6ST-03	18.36	5.11	723.91			

**Table 2 polymers-16-01715-t002:** Static load tensile test results at low temperature.

Specimen Number	Maximum Failure Load (KN)	Maximum Load Displacement (mm)	Tensile Strength (MPa)	Average Failure Load (KN)	Average Load Displacement (mm)	Average Strength (MPa)
B2DT-01	10.04	3.06	785.38			
B2DT-02	10.09	3.05	796.81	10.09	3.06	789.94
B2DT-03	10.13	3.08	787.63			
B2ST-01	14.26	2.91	561.32			
B2ST-02	14.40	2.94	576.92	14.37	2.93	569.14
B2ST-03	14.46	2.93	569.19			
B4DT-01	10.48	3.75	830.36			
B4DT-02	10.52	3.73	820.43	10.53	3.75	824.71
B4DT-03	10.59	3.76	823.33			
B4ST-01	17.32	3.05	674.25			
B4ST-02	17.39	3.03	695.56	17.38	3.03	683.98
B4ST-03	17.44	3.01	682.12			
B6DT-01	11.30	4.93	892.36			
B6DT-02	11.35	4.92	889.33	11.34	4.94	892.53
B6DT-03	11.38	4.96	895.91			
B6ST-01	19.42	4.52	763.17			
B6ST-02	19.45	4.56	769.26	19.46	4.56	765.95
B6ST-03	19.51	4.61	765.43			

**Table 3 polymers-16-01715-t003:** Damage morphology of the specimens after the static load tensile tests.

Room Temperature	Low Temperature
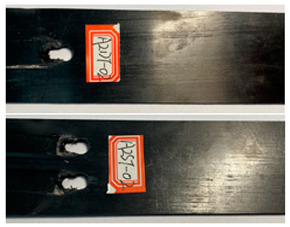	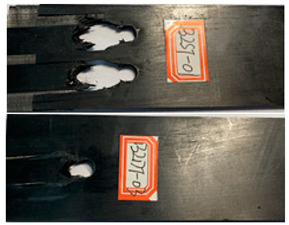
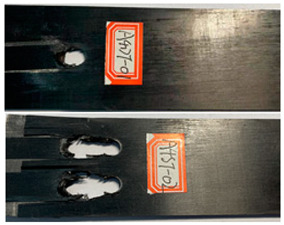	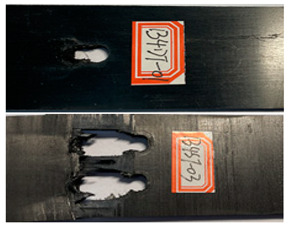
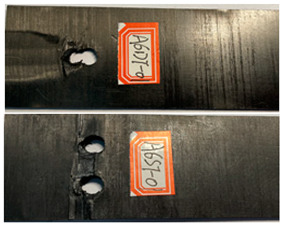	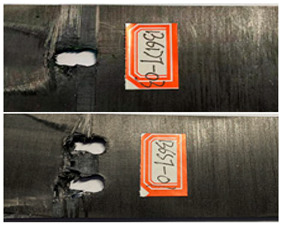

**Table 4 polymers-16-01715-t004:** Tensile fatigue test results at room temperature.

Stress Level	Specimen Number	Maximum Load (KN)	Minimum Load (KN)	Fatigue Stress (MPa)	Fatigue Life	Average Fatigue Life	Standard Deviation
85%	A2SV-01	11.36	1.14	446.43	1184	1364	
	A2SV-02			443.58	1366		146.1597
	A2SV-03			447.17	1542		
80%	A2SV-04	10.69	1.07	414.21	18,663	20,761	
	A2SV-05			418.81	21,080		1599.142
	A2SV-06			422.71	22,541		
75%	A2SV-07	10.02	1.00	395.08	254,008	261,445	
	A2SV-08			396.96	263,556		5420.073
	A2SV-09			391.26	266,771		
85%	A6SV-01	15.56	1.56	610.46	1986	2287	
	A6SV-02			597.13	2321		232.7464
	A6SV-03			604.58	2553		
80%	A6SV-04	14.65	1.47	568.42	24,963	26,646	
	A6SV-05			581.22	26,725		1342.676
	A6SV-06			565.94	28,249		
75%	A6SV-07	13.73	1.37	525.17	333,215	355,775	
	A6SV-08			540.46	364,130		16,130.41
	A6SV-09			530.24	369,981		

**Table 5 polymers-16-01715-t005:** Tensile fatigue test results data at low temperature.

Stress Level	Specimen Number	Maximum Load (KN)	Minimum Load (KN)	Fatigue Stress (MPa)	Fatigue Life	Average Fatigue Life	Standard Deviation
85%	B2SV-01	12.21	1.22	478.36	1735	1859	
	B2SV-02			481.43	1864		99.26731587
	B2SV-03			478.24	1978		
80%	B2SV-04	11.50	1.15	448.43	26,149	27,522	
	B2SV-05			451.29	27,783		1031.506773
	B2SV-06			449.05	28,635		
75%	B2SV-07	10.78	1.08	414.94	336,879	360,330	
	B2SV-08			425.65	364,130		17,800.66014
	B2SV-09			425.04	379,982		
85%	B6SV-01	16.54	1.65	646.92	2784	2919	
	B6SV-02			647.84	2941		102.4337184
	B6SV-03			651.07	3032		
80%	B6SV-04	15.57	1.56	603.29	36,445	37,545	
	B6SV-05			615.80	37,258		1035.843081
	B6SV-06			602.13	38,933		
75%	B6SV-07	14.60	1.46	572.80	584,340	612,520	
	B6SV-08			566.48	596,603		31,580.16521
	B6SV-09			578.40	656,616		

**Table 6 polymers-16-01715-t006:** Damage morphology of the specimens after the static load fatigue tests.

Stress Level	Room Temperature	Low Temperature
85%	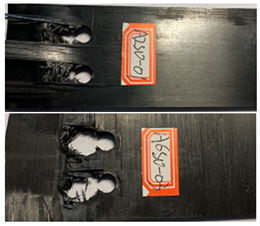	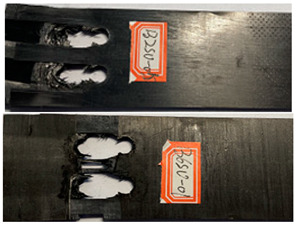
80%	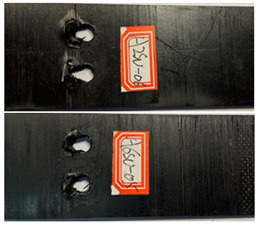	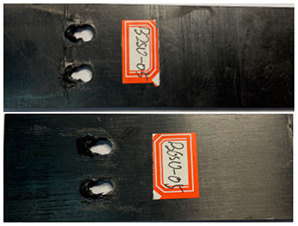
75%	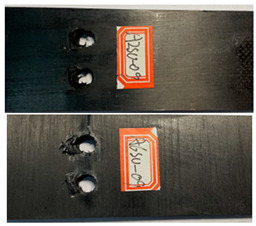	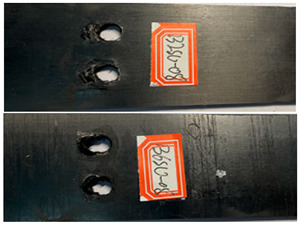

## Data Availability

All the results of this study can be found in the paper.

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
