# Peer review of "Mechanical Properties of Carbon Fiber-Reinforced Plastic with Two Types of Bolted Connections at Low Temperatures"

_polymers, 2024, doi:10.3390/polym16121715_

Round 1

Reviewer 1 Report

Comments and Suggestions for Authors

Abstract:

- Define "low room temperatures" numerically for clarity.

- Specify "different tightening torques" with exact values.

- Clarify "great specific strength, specific modulus" quantitatively.

Section 1:

- Detail "different variables" tested in joint strength.

- Provide numeric values for "low temperature" effects.

- Cite sources for "preload force" effect on joint failure.

- Clarify "without damaging the material" methodology.

- Consider citing https://doi.org/10.1016/j.promfg.2020.04.107 on pultruded profiles that are also widely used in the aerospace industry.

Section 2:

- Clarify measurement accuracy for temperature control in autoclave.

- Detail criteria for selecting higher strength bolt connections.

- Define "significant permanent deformation" quantitatively.

- Expand on how preload force affects connection stability.

- Contrast findings with existing CFRP bolt connection research.

Section 3:

- Clarify "abrupt inflection points" with exact load values.

- Quantify "compression force" in terms of tightening torque.

- Detail "decrease of internal thermal stress" numerically.

- Explain variance in failure loads across temperatures.

- Provide statistical significance of tensile test results.

Section 4:

- Elaborate on how temperature influences structural integrity.

- Clarify criteria for "obvious failure" at the connection hole.

- Describe implications of shear failure for practical applications.

- Discuss potential scalability of findings to other composite types.

- Recommend future studies on long-term durability under varied torques.

Reviewer 2 Report

Comments and Suggestions for Authors

Reviewer comments

Manuscript ID: polymers-3014813

Title: Mechanical properties of Carbon Fiber Reinforced Plastic with two types of bolted connection at low temperature

Journal: Polymers

The manuscript focuses on the study of the mechanical properties of bolted joints made of carbon fiber reinforced plastic (CFRP) subjected to both static and fatigue tensile tests.While the paper is well-aligned with the thematic scope of the journal, there are areas that require improvement. Therefore, I suggest the following revisions to enhance the quality and depth of the manuscript:

1. The main objective of this work should be clearly addressed in the abstract

2. The novelty of this paper is not well described in the introduction. The novelty should be clearly addressed, highlighting the findings, or improvements compared to similar research previously published.

3. The value of bolt joint tightening torque should be indicated clearly. Is there any discussion about the effect of an excessive tightening torque on the damage initiation and compression failure?. Add figure if possible to clearly indicate this effect.

4. Standard deviations for experimental tests should be added in tables and figures.

5. In Fig. 8 (S-N curve): how the maximum fatigue stress is determined. Authors should present the variations of the maximum load during fatigue tests.

6. Authors indicate that “the load coefficients were 85%, 80% and 75% of the failure strength” why this values are chosen, are not an excessive loading?

7. the authors claim in the conclusion that “With the increase of tightening torque and the decrease of temperature, the static load tensile mechanical properties of CFRP bolt structure are improved”. Limit value of tightening torque?

8. It advises making the conclusion more succinct by quantifying improvements and focusing on highlighting the primary results of the study.

9. Typos and grammar problems need to be corrected properly, authors should carefully check through the manuscript before submitting a revision.

The reviewer recommends that the author do major revision to the manuscript.

Comments on the Quality of English Language

Typos and grammar problems need to be corrected properly, authors should carefully check through the manuscript before submitting a revision.

Reviewer 3 Report

Comments and Suggestions for Authors

In the present research work, a facile and efficient method to prepare carbon fiber reinforced plastic (CFRP) bolted laminates was proposed and subsequently specimens were tested for mechanical performance. CRRP specimens of two bolted connections, i.e. single-nailed and double-nailed, were characterized for static load tensile and tensile fatigue properties. It was found that the tensile strength of the double bolted specimens was more than 20% higher than that of the single-stud specimens. The failure curves and fatigue life relationships of specimens were obtained with two different connection methods at different tightening torques and low room temperatures. Moreover, the analysis revealed the effect of bolt tightening torque and temperature on the structural mechanical properties of bolted CFPR. Finally, it was claimed that the present research work provided data reference for researchers to design the composite bolted joint structure under the airplane flight environment.

Suggestions/Concerns/Issues

“…….because of its great specific strength, specific modulus and so on.” AND “……such as connection form, connection geometry, connected environment and so on.” Be specific instead of so on.

Is there any specific reason for using unidirectional fiber prepregs and then laying these in two directions than opting for two-directionally woven fabric?

Why did you use 16 layers of prepregs? Is there any specific requirement of thickness or specimens?

For the tensile fatigue test, load coefficients were 85%, 80% and 75% of the tensile strength while frequency was 10 Hz. Why did you choose these parameters? Kindly explain.

“The laminate was heated from RT (25 °C) to 120 °C at a steady rate (1-2 °C/min) and then held at 120 °C for 90 minutes according to equipment recommendations.” Was it epoxy curing requirement or equipment recommendations?

Was the thickness of specimens 2.10 mm. If yes, please mention in the text of the manuscript?

Is there any mutual relationship between thickness of specimen, width of specimen, location of holes and diameter of holes or have you simply followed a standard (ASTM-3039)?

Did you use any tabs for easy gripping of the specimens’ ends in the jaws of machine or not?

“….hole is more than 1.5 times the diameter of the screw hole.” Is it standard practice?

The number XPJL-ZZ seems difficult to understand unless it is standard practice. For example: when J is D, it means single nail and when J is S, it means double nail connection. Although S for single and D for double is easy to comprehend. Similarly, T can stand for tensile while F can stand for Fatigue test.

Please give numbers to each curve in Figure 6 for easy understanding of the readers. Nevertheless, it is up to you.

Tables 3 and 6 should be Figures. Can you please include the high magnification images of the holes after testing to understand the mode of fracture easily.

Just a general question! Did you have any doubt before the experiments that the double-stud bolt connection was not stronger than the single stud?

Provide standard errors with the values given in Table 4, Table 5 and Figure 7.

Do you think that by simply observing the SEM image (Figure 9), one can conclude that bon between resin and fiber is strong or weak? A qualitative observation may be provided but unless you quantify it, it cannot be concluded that bonding at low temperature is better than at normal temperature. If you are still positive, please provide a high magnification image.

Mention operating parameters of SEM (Figure 9)

Comments on the Quality of English Language

Moderate editing of English language required

Round 2

Reviewer 1 Report

Comments and Suggestions for Authors

My main concerns were addressed. The paper can be accepted as is.

Author Response

Thank you for the valuable comments about the revision.

Reviewer 2 Report

Comments and Suggestions for Authors

The authors have provided correct responses to the questions raised by the reviewer. In light of the revisions carried out, I recommend that the manuscript be accepted for publication.

Comments on the Quality of English Language

in the abstract: "CFRP bolted" and not "CFPR bolted"

Author Response

(The authors gave the same response as above.)
